# Response Surface Methodology Optimization in High-Performance Solid-State Supercapattery Cells Using NiCo_2_S_4_–Graphene Hybrids

**DOI:** 10.3390/molecules27206867

**Published:** 2022-10-13

**Authors:** Zhong-Yun Hong, Lung-Chuan Chen, Yu-Chu M. Li, Hao-Lin Hsu, Chao-Ming Huang

**Affiliations:** 1Department of Materials Engineering, Kun Shan University, Tainan 710, Taiwan; 2Department of Mechanical Engineering, Southern Taiwan University of Science and Technology, Tainan 710, Taiwan; 3Green Energy Technology Research Center, Kun Shan University, Tainan 710, Taiwan

**Keywords:** NiCo_2_S_4_, graphene, response surface methodology, optimization, supercapattery cell

## Abstract

In this work, NiCo_2_S_4_–graphene hybrids (NCS@G) with high electrochemical performance were prepared using a hydrothermal method. The response surface methodology (RSM), along with a central composite design (CCD), was used to investigate the effect of independent variables (G/NCS, hydrothermal time, and S/Ni) on the specific capacitances of the NCS@G/Ni composite electrodes. RSM analysis revealed that the developed quadratic model with regression coefficient values of more than 0.95 could be well adapted to represent experimental results. Optimized preparation conditions for NCS@G were G/NCS = 6.0%, hydrothermal time = 10.0, and S/Ni = 6.0 of NCS@G (111) sample. The maximum specific capacitance of NCS@G (111)/Ni fabricated at the optimal condition is about 216% higher than the best result obtained using the conventional experimental method. The enhanced capacitive performance of the NCS@G (111) sample can be attributed to the synergistic effect between NCS nanoparticles and graphene, which has the meso/macropores conductive network and low diffusion resistance. Notably, the NCS@G (111) could not only provide numerous reaction sites but also prevent the restacking of graphene layers. Furthermore, a supercapattery cell was fabricated with an (G + AC)/Ni anode, a NCS@G (111)/Ni cathode, and a carboxymethyl cellulose–potassium hydroxide (CMC-KOH) gel electrolyte. The NCS@G (111)//(G + AC) demonstrates an outstanding energy density of 80 Wh kg^−1^ at a power density of 4 kW kg^−1^, and a good cycling performance of 75% after 5000 cycles at 2 A g^−1^. Applying the synthesis strategy of RSM endows remarkable capacitive performance of the hybrid materials, providing an economical pathway to design promising composite electrode material and fabricate high-performance energy storage devices.

## 1. Introduction

Global warming, resulting from greenhouse gas emissions generated from the combustion processes for fossil fuel, has brought severe destruction to global ecosystems and threatens human beings. One of the solutions to reduce greenhouse gas emissions is the replacement of fossil fuels with renewable energies. Holechek [1] reported that the replacement of fossil fuels with renewable energy by 2050 could be possible under some circumstances, such as aggressive application of all eight pathways, lifestyle changes in developed countries, and close cooperation among all nations. However, renewable energy produced by solar or wind is intermittent and needs to be stored instantly by electrochemical energy-storage (EES) devices if it is to be a reliable energy supply. EES devices are essential for developing an efficient and clean energy system, and the booming construction of renewable energies has promoted the rapid development of EES devices, e.g., lithium-ion batteries (LIBs) and supercapacitors (SCs). For example, in solar photovoltaic energy storage applications, SCs can provide frequency regulation, while LIBs meet the slower-changing demand requirement [2]. LIBs have several disadvantages, such as shorter cycle lives and longer charging times; it is especially urgent that we resolve the safety concerns resulting from the use of flammable organic electrolytes. Compared with LIBs, supercapacitors have improved safety, short charge/discharge times, long cycle lives, and high power density. However, presently, SCs still have a low energy density of about 40–90 Wh/kg [3], compared to about 150–250 Wh/kg for LIBs [4]. The usage of SCs in hybrid electric vehicles and renewable energy storage grids will be greatly enhanced if the energy density of SCs can be significantly improved. In 2018, Chen et al. [5] proposed a high-performance supercapattery device with the advantages of combining the high energy density of batteries and the high power density of SC. The supercapattery has a positive electrode made up of battery-type materials to provide energy density, and the negative electrode consists of carbon-based materials to give high power density. Recently, ternary Ni-Co sulfides with rich redox-active sites and variable valence states, such as Ni^2+^/Ni^3+^, Co^2+^/Co^3+^, and Co^3+^/Co^4+^, during the charging and discharging process have been suggested as promising battery-type materials for supercapatteries; however, the low conductivity, easy agglomeration, and poor electrochemical stability of ternary Ni-Co sulfides present a series of problems to be tackled [6,7]. Many results have reported that the combination of high conductivity materials, such as reduced graphene oxide (rGO) or graphene (G) and Ni-Co sulfides using a hydrothermal process, can achieve a synergistic effect, leading to enhanced electrochemical performance [8,9,10,11,12]. Dong et al. [9] prepared Ni_x_Co_y_S_4_ nanosheets (various Ni-Co ratios) on rGO using a hydrothermal method, and found that the Ni_1.64_Co_2.40_S_4_/rGO electrode delivers a high specific capacitance of 1089 F g^−1^ at 1 A g^−1^. He et al. [12] synthesized the NCS/G hybrid using the hydrothermal process to investigate the effect of amounts of thioacetamide on the electrochemical performance; they showed that graphene effectively enhances the structural stability of the as-prepared NCS/G. Traditional hydrothermal approaches for the preparation of NCS@G composite materials are dependent on the change of one independent variable parameter, such as Ni-Co or Ni-S ratios, Ni-Co sulfides g ratio, and hydrothermal time, while keeping all other variables constant. Thus, traditional approaches result in the additional consumption of chemicals and increased time and costs; moreover, conventional techniques cannot illuminate the synergistic effect of the process parameters. To solve this issue, we need a new strategy with a highly efficient approach for the improved development of NCS@G hybrid composites. 

Recently, there have been many reports employing the response surface methodology (RSM) and central composite design (CCD) to identify the effective parameters for reducing the number of experimental runs and to determine the optimal operating parameters of the system [13,14,15]. Nowadays, RSM is the most popular method for optimization processes, as it provides the details of the interaction and quadratic effects of the process variables. So far, no literature has used the RSM in the hydrothermal process for the optimization of the NCS@G hybrid composite used as electrode material for supercapacitors. In this study, an efficient investigation of NCS@G hybrid composite synthesis was carried out by optimizing and varying parameters such as G/NCS ratio, hydrothermal time, and S/Ni ratio, taking into account their interactions and effects on each condition. The modeling was conducted using Design Expert Software and RSM with CCD. In our study, a G/NCS ratio of 6%, a hydrothermal time of 10 h, and a S/Ni ratio of 6 were the optimal conditions for reaching a maximum of 2380 F g^−1^.

## 2. Results and Discussion

### 2.1. The Hydrothermal Synthesis of NCS@G Hybrids Using the Traditional Method

The hydrothermal synthesis of the NCS@G hybrid was studied for various graphene additions, hydrothermal times (T), and ratios of S/Ni. The reaction scheme, developed through a one-step hydrothermal approach, is shown as follows (Figure 1).

For the formation of NCS@G, graphene paste was first dispersed in H_2_O/EG. After adding Ni(NO_3_)_2_·6H_2_O and Co(NO_3_)_2_·6H_2_O, the Ni^2+^ and Co^2+^ ions were adsorbed on the graphene particles. Experiments were performed by varying one factor simultaneously, with the remaining factors kept constant. In the beginning, five NCS@G samples were prepared, changing the graphene paste to NCS powder of five amounts of addition, by keeping the hydrothermal time of 6 h and the thiourea to Ni(NO_3_)_2_·6H_2_O (S/Ni) of 4.0. Figure 1a shows the effect of the ratio of G/NCS on the specific capacitance, which increases as the G/NCS ratio increases up to 4%, and then decreases. Further increasing the G/NCS does not favor electrical conductivity, which might be due to an agglomeration between graphene particles. Then, the effect of hydrothermal times on the specific capacitance was studied. A similar trend of maximum specific capacitance was obtained at the hydrothermal time of 8 h while G/NCS and the ratio of S/Ni are held constant at 4% and 4, respectively, as shown in Figure 1b. Finally, Figure 1c reveals that the specific capacitance initially increases with the ratio of S/Ni and then decreases significantly for S/Ni ratios of over 5.0. In comparison, the G/NCS ratio and hydrothermal time are held constant at 4% and 6 h, respectively. The maximum specific capacitance, 1100 F g^−1^, was obtained at a G/NCS of 4.0%, a hydrothermal time of 8 h, and an S/Ni ratio of 5.0. This result is the best yet obtained using the traditional experimental method.

### 2.2. The Improvement of Specific Capacitance of NCS@G/Ni Composite Electrodes with CCD

To improve the specific capacitance in hybrids, RSM was used to optimize the level of independent variables. The effects of independent variables (G/NCS (X_1_), hydrothermal time (X_2_), and S/Ni (X_3_) on the response (the specific capacitances of the NCS@G/Ni composite electrodes at 5 A g^−^^1^) are given in Table 1. The regression equation for the response variable, obtained from the response surface methodology, is stated by a quadratic equation (Equation (1)):Specific capacitance = +12171.8 − 360.7 X_1_ − 1322.8 X_2_ − 2434.8 X_3_ + 11.3 X_1_X_2_ + 70.3 X_1_X_3_ + 72.3 X_2_X_3_ + 16.8 X_1_^2^ + 57.4 X_2_^2^ + 173.3 X_3_^2^(1)

The analysis of variance (ANOVA) test is an important tool to determine whether there are any statistically significant differences between the means of three or more independent groups. The ANOVA of the quadratic surface modeling for the specific capacitances of the NCS@G/Ni composite electrodes is provided in Table 2. The significance of the variables is indicated by the *F* and *p* values, where the values of *p*-value < 0.05 show that the model terms are significant. The more significant effect of the corresponding variable is present if there is a larger F-value and a smaller *p*-value [16]. As shown in Table 2, the *p*-value < 0.0001 and F-value at 49.71 for specific capacitances implies that the model is significant. G/NCS (*X*_1_) and S/Ni (*X*_3_) were found to significantly affect specific capacitance, with G/NCS having the greatest effect. A *p*-value of 0.7230 showed that ‘‘Lack of Fit’’ was insignificant, and the regression model was effective for the specific capacitances of NCS@G/Ni composite electrodes. The ANOVA table shows that the G/NCS (*X*_1_), S/Ni (*X*_3_), interactions of G/NCS - S/Ni (*X*_1_X_3_), hydrothermal time - S/Ni (*X*_2_X_3_), and quadratic term of hydrothermal time (*X*_2_^2^) and S/Ni (*X*_3_^2^) were significant to the specific capacitance based on *p*-values of less than 0.05. Moreover, ANOVA indicated that the coefficient of determination (R^2^) and adjusted R^2^ were 0.9781 and 0.9585, respectively, which indicates a stronger association between the factors and predicted results. Hence, the model can be reasonably adapted to analyze the responses. Since *X*_1_*X*_3_, *X*_2_*X*_3_, *X*_2_^2^, and *X*_3_^2^ are important terms in the model, the traditional experimental method cannot accurately describe the capacitance behavior.

Figure 2 shows the plots of residuals against the normal probability for specific capacitance. As shown in the plot, the obtained data show a linear relationship, which indicates the distributed normal distribution of the residuals. Thus, it is concluded that the prediction of the experimental results obtained from the developed quadratic model is satisfactory in the parameterization of the specific capacitance.

Numerical optimization was executed using the desirability function of Design Expert Software; the goal was to find the maximum specific capacitance within the independent variables of G/NCS, hydrothermal time, and S/Ni. The solution with maximum specific capacitance was selected as the optimized preparation of the NCS@G/Ni composite electrode condition. The two-dimensional (2D) plots in Figure 3 represented Equation (1) and were used to predict the specific capacitance for various values of independent variables. The maximum specific capacitance is 2317 F g^−1^ at G/NCS = 6.0%, hydrothermal time = 10.0, and S/Ni = 6.0.

Figure 4 shows the 3D response surfaces constructed to show the effects of the two most important variables (G/NCS and S/Ni) on specific capacitance. The figure demonstrates that specific capacitance increases with an increase in the G/NCS and S/Ni at the hydrothermal time of 10 h.

The specific capacitances of the NCS@G/Ni composite electrodes under the optimal conditions were performed to confirm the improvement achieved using the statistically designed experiments. The optimized preparation conditions were repeated three times to check the model’s accuracy for the response value prediction, and the results are shown in Table 3. The response value at the optimized preparation conditions was at G/NCS = 6.0%, hydrothermal time = 10.0, and S/Ni = 6.0 (Figure 3). The average specific capacitance of optimally conditioned composite electrodes, NCS@G (111)/Ni, at three runs is 2376 F g^−1^ at 5 A g^−^^1^, which is in good agreement with the predicted value (2317 F g^−1^), shown in Figure 3. Moreover, the maximum specific capacitance of NCS@G (111)/Ni fabricated at the optimal conditions is about 216% higher than the best result obtained using the conventional experimental method, which occurred at a G/NCS of 4.0%, a hydrothermal time of 8 h, and an S/Ni of 5.0. This similarity between the predicted and observed results reflects the accuracy and applicability of the CCD for optimizing the synthesis of the NCS@G hybrid.

### 2.3. Structural, Morphological, and Textural Analysis of the NCS@G Hybrids

Figure 5 shows the crystal structures of the synthesized NCS, the hybrid NCS@G (NCS@G (000), and NCS@G (111)), characterized by powder X-ray diffraction. For pure NCS, the peaks at 2θ values of 26.7°, 31.8°, 38.2°, 50.4°, and 55.3° can be indexed to the (220), (311), (400), (511), and (440) planes of cubic phase NiCo_2_S_4_ (JCPDS 43-1477). The hybrid samples with different graphene-loaded samples (NCS@G (000) and NCS@G (111)) show a similar XRD pattern as that of the pure cubic NiCo_2_S_4_. No extra peak or shift in the peak position was observed, suggesting that the presence of graphene does not influence the crystal structure of hybrid NCS@G during the synthesis process.

Raman spectroscopy was used to evaluate the structures of the graphene-based materials. The results are shown in Figure 6. Three peaks at 190, 516, and 653 cm^−1^ were observed for NCS and NCS@G hybrids, whereas the peaks at 516 and 653 cm^−1^ belong to the F2g and A1g modes of NiCo_2_S_4_, respectively [7]. These results confirmed the formation of NiCo_2_S_4_ again through the hydrothermal syntheses of NCS, NCS@G (000), and NCS@G (111). For the NCS@G (000) and NCS@G (111), extra peaks of D (1347), G (1580), and 2D band (2600) cm^−1^ belonging to graphene were observed. The D to G band integrated intensity ratio reflects the degree of the disordered structures to the graphitization of graphene. The integrated intensity ratios of I(D)/I(G) are calculated to be 1.10 for the NCS@G (111) and 1.07 for NCS@G (000), revealing the existence of highly disordered graphene structures. The values of I(D)/I(G) of NCS@G (111) and NCS@G (000) are close, due to the usage of the same graphene paste.

Figure 7 depicted the morphology of the NCS, NCS@G (000), and NCS@G (111). For pure NCS, rounded microspheres with a diameter of about 800–900 nm were observed. The images were quite different for NCS@G hybrids. The morphology of NCS@G (000) is shown in Figure 7b, and the morphology of NCS is still observed. However, the morphology was changed to nanoparticles, demonstrating the presence of ~50 nm-sized NCS nanoparticles on the graphene surface due to the binding effect of graphene. Figure 7c shows the morphology of NCS@G (111); it displays the graphene sheets covered with NCS nanoparticles in some places, and the graphene sheets with fewer amounts of NCS loaded are seen in the image. The layered structure of graphene is observed because of NCS nanoparticles which prevent the restacking of graphene layers. Consequently, the electrochemical properties of NCS@G hybrids could be enhanced due to the increased reaction sites of the synergistic effect of graphene and NCS nanoparticles.

The textural properties of NCS, NCS@G (000), and NCS@G (111) could significantly affect the electrochemical performance of composite electrodes. Figure 8 shows nitrogen adsorption–desorption isotherms of NCS, NCS@G (000), and NCS@G (111); the corresponding pore size distributions are also displayed in the inset. The non-porous structure of NCS is confirmed, since a horizontal line of low adsorbed N_2_ volume was observed. After the addition of graphene, the isotherm curves of NCS@G (000) and NCS@G (111) were quite different from the NCS. For NCS@G (000), the relative pressure, *P/P0*, ranges from 0.9 to 1.0; for nitrogen gas, the adsorption volume increases sharply, illustrating that the NCS@G (000) contains many macropores. The N_2_ adsorption/desorption curves of NCS@G (111) show a hysteresis loop and obvious type Ⅳ adsorption when the relative pressure is ranged from 0.45 to 1.0, which manifests in the presence of mesopores. A steep increase at a high relative pressure (*P/P0* > 0.9) without an adsorption plateau was observed, suggesting the existence of macropores. The specific surface areas were 1.1, 14.0, and 19.7 m^2^ g^−1^ for NCS, NCS@G (000), and NCS@G (111), respectively, and the average pore size was 3.8, 22.2, and 36.0 nm for NCS, NCS@G (000), and NCS@G (111), respectively. It is obvious that the presence of graphene can effectively increase the porosity and specific surface area, which is expected to enhance the capacitance. SEM and textural properties confirmed that the NCS powders could act as spacers, preventing the aggregation of graphene and further resulting in the conductive network of meso/macropores.

### 2.4. Electrochemical Performance of the NCS, NCS@G (000), and NCS@G (111) Electrodes

Figure 9 compares the GCD curves of the as-prepared NCS, NCS@G (000), and NCS@G (111) electrodes in 6 M KOH between 0–0.5 V at the current density of 5 A g^−1^. A little deviation from the symmetric triangular curve of NCS was observed, indicating rapid charge–discharge rates. It is clear that the NCS@G (111) electrodes exhibit a nonlinear shape with battery-like characteristics, and a discharge plateau with the voltage around 0.35 V (vs. SCE), signifying a typical noncapacitive faradaic reaction. The order of specific capacitance for the three electrodes is NCS@G (111) (2380 F g^−1^) > NCS@G (000) (1040 F g^−1^) > NCS (819 F g^−1^). The outstanding specific capacitance of NCS@G (111) results from the appropriate addition of graphene and NCS nanoparticles, due to their conductive 3D networks and porous structures.

### 2.5. Electrochemical Impedance of the Electrode

Figure 10 presents the impedance spectra of NCS, NCS@G (000), and NCS@G (111) electrodes with an enlarged view (inset i) and a fitted equivalent circuit (inset ii). In the high-frequency region, the intercept with the *x*-axis is the ohmic resistance derived from the internal resistance of the electrolyte, the contact resistance between the electrolyte and the electrode, and the intrinsic resistance of the electrode (*R_s_*). The estimated *R_s_* values were 0.60, 0.51, and 0.72 Ω for the NCS/Ni, NCS@G (000)/Ni, and NCS@G (111)/Ni electrodes, respectively. In the medium-high-frequency region, the semicircle diameter region corresponds to a parallel combination of charge transfer resistance (*R_ct_*) and the double-layer capacitance (C_d_) on the grain surface, where *R_ct_* stands for the charge-transfer resistance at the interface between the electrolyte and electrode. As shown in inset (i), a skinny semicircular arc of NCS/Ni was observed, which might be due to a microsphere structure displaying low charge-transfer kinetics. In contrast, due to the addition of graphene, no semicircles were observed, indicating the lower faradic charge-transfer resistance of the NCS@G (000)/Ni and NCS@G (111)/Ni electrodes. In the low-frequency region, the straight line is the Warburg resistance (*W_s_*), responding to the ion diffusion/transport resistance from an electrolyte to the surface of the electrode. The high slope (>45°) of the Warburg straight lines indicates that the diffusion resistance of the electrolyte ions in the NCS@G (111)/Ni electrode is lower than those of the NCS/Ni and NCS@G (000)/Ni electrodes. The high specific capacitance and low resistance of the NCS@G (111)/Ni electrode may be attributed to the following reasons: (1) the meso/macropores structure provides numerous open reservoirs that can enhance the mass transport of electrons for rapid redox reactions and significantly improve the electrolyte/electrode contact area, increasing charge storage reactions; (2) the synergistic effect from NCS and graphene improves both the capacitive Faradaic (pseudocapacitive) and capacitive non-Faradaic (EDLC).

### 2.6. Assembly and Performance of a Solid-State Supercapattery Cell

It has been reported that a supercapattery cell can be assembled of a battery-type electrode and a capacitor-type electrode to exhibit two different potential windows in the same electrolyte [17,18]. Using such a design, the energy density of the supercapattery cell is significantly enhanced, due to the widened operation voltage window. As such, a supercapattery cell was fabricated using the highest performance electrode, NCS@G (111)/Ni, as a positive electrode, and (G + AC)/Ni in 3.5 × 7 cm^2^ as the negative electrode in the CMC-KOH gel electrolyte. A nickel strip wired both the positive electrodes and negative electrodes for battery testing. After that, the supercapattery cell NCS@G (111)//(G + AC) was sealed by hot-pressing two pieces of Al-plastic films.

Figure 11a shows the CV curves of an optimized NCS@G (111)//(G + AC), operated from 1.0 to 1.6 V at a scan rate of 50 mV s^−1^. When the potential window expanded to 1.6 V, the CV’s shape remained analogous, without obvious distortion. The CV studies show that, for NCS@G (111)//(G + AC), 1.6 V is a suitable potential window; therefore, the electrochemical performance of the as-fabricated cell was carried out in the same potential window. The rate-dependent NCS@G (111)//(G + AC) at various scan rates from 5 to 100 mV s^−1^ is shown in Figure 11b. The nature of the CV plots indicated no obvious changes with increasing scan rates, and kept relatively quasi-rectangular profiles at the high scan rate of 100 mV s^−1^. Due to appropriate graphene acting as the electronic conductive channel to increase electrical conductivity, the low contact resistance of the NCS@G (111)/Ni electrodes, as shown in Figure 10, results in the distinguished rate capability of the supercapattery cell. The GCD plots of the supercapattery cell at various current densities of 2, 5, 8, 10, and 20 A g^−^^1^, with an applied window of 0–1.6 V, are shown in Figure 11c. The pseudocapacitive behavior was observed for GCD plots at various current densities, and no obvious IR drop was observed in the discharge curves. Based on the GCD curves, the specific capacitance of the supercapattery cell was calculated, and the mass-specific capacitances of the cell at 2, 5, 8, 10, and 20 A g^−1^ were 279, 224, 192, 173, and 130 F g^−1^, respectively, where the total mass of the active materials in both electrodes was about 3.24 mg cm^−2^. The cycling test of the NCS@G (111)//(G + AC) was characterized at a current density of 2 A g^−1^ between 0 and 1.6 V for 5000 charge/discharge cycles, as displayed in Figure 11d. The cell showed superior electrochemical stability with 100–123% of the initial capacity retention during the period of 1 to 3100 cycles. The capacitance still maintained 75% of the initial capacitance after 5000 cycles. The observed positive capacity retention could be attributed to further activation of the NCS@G (111)/Ni electrode during the charging/discharging cycles [19].

Figure 12 shows the EIS measurements before and after the charge/discharge for 5000 cycles. As shown in the inset, the respective estimated *R_s_* values were 0.65 and 0.47 Ω for the after and before stability tests, respectively. For the *R_ct_*, as shown, the semicircle is significant and large, thus signifying that the faradaic charge transfer resistance became high after 5000 cycles. Moreover, the slope of the straight line (Warburg resistance (*W_s_*)) of the supercapattery cell after the 5000 th cycle is much lower than that of the first cycle. This observation indicates that the 5000 th cycle of the cell has much higher diffusive resistance than the first cycle, indicating the electrolyte ions’ decreased diffusion and migration pathways during the charge/discharge processes. The increased *R_ct_* and *W_s_* probably resulted from the loss of adhesion of some NCS@G (111) material with the current collector or the dissolution of NCS@G hybrids during the charge/discharge cycling.

The energy density (*E*) and power density (*P*) of the assembled NCS@G (111)//(G + AC) were calculated based on the GCD curves using Equations (7) and (8), and the Ragone plots of the supercapattery cell are shown in Figure 13. It is worth noting that the maximum energy density obtained for NCS@G (111)//(G + AC) was 80 Wh kg^−1^ at a power density of 4000 W kg^−1^ and with a voltage window of 1.6 V. These values are higher than other NiCo_2_S_4_@rGO or NiCo_2_S_4_@graphene composites hybrid supercapacitors, such as NiCo_2_S_4_//G@CNT (45 Wh kg^−1^ at 800 W kg^−1^) [11], NiCo_2_S_4_@rGO//AC (42 Wh kg^−1^ at 1067 W kg^−1^) [20], NiCo_2_S_4_@EEG//NMCS (30 Wh kg^−1^ at 800 W kg^−1^) [21], NiCo_2_S_4_@rGO//N g (34 Wh kg^−1^ at 411 W kg^−1^) [22], and NiCo_2_S_4_@rGO//rGO (47 Wh kg^−1^ at 1200 W kg^−1^) [23].

The superior electrochemical performances of the NCS@G (111)//(G + AC) supercapattery cell can be attributed to three factors. First, the presence of graphene could effectively increase the porosity and specific surface area, providing more numerous reaction sites than pure NCS and graphene, which is beneficial for the migration of electrons and ions. Second, the synergistic effect between NCS and graphene results in the formation of NCS nanoparticles and the prevention of graphene sheet stacking. Third, the gel electrolyte allows the charge to accumulate efficiently due to the presence of rich sites, improving ion transportation and extending the operating voltage.

## 3. Materials and Methods

### 3.1. Materials

Analytical-grade nickel nitrate hexahydrate, cobalt nitrate hexahydrate, thiourea, and PVDF were purchased from (Sigma, St. Louis, USA, >99%). The graphene pastes (TCMC, Taoyuan, Taiwan, E-Closer^®^ 040 WB, 4% active content) were used as received. Nickel foam (UBIQ, Taoyuan, Taiwan, 110 pores per square inch) was used as the substrate of the electrode. Mesoporous activated carbon (CSCC, Taiwan, ACS25, BET 2500 ± 200 m^2^ g^−1^) was used without further treatment.

### 3.2. Synthesis of the NCS@G Hybrids

In a typical one-step hydrothermal procedure for preparing the NCS@G(000), 3.17 g of the graphene paste was dispersed into 50 mL of H2O/ethylene glycol solvent (1:1 by volume) through ultrasonication for 0.5 h to form a homogeneous graphene suspension. Then, 1 mmol of Ni(NO_3_)_2_·6H_2_O and 2 mmol of Co(NO_3_)_2_·6H_2_O were gradually added into the above suspension and stirred for 0.5 h. Subsequently, 5 mmol of thiourea was introduced into the above solution with the ratio of Ni(NO_3_)_2_·6H_2_O:Co(NO_3_)_2_·6H_2_O:thiourea as 1:2:5. The resultant mixture was stirred for 1 h. Then, the solution was transferred into a 200 mL Teflon-lined stainless-steel autoclave; the reaction was carried out at 210 °C for 8 h and then cooled down to room temperature. Finally, the as-obtained samples were collected by filtration, washed with DI water and ethanol mixture several times, and dried in the vacuum oven for 12 h at 100 ℃. In the case of NCS@G(111), 4.76 g of the graphene paste, a hydrothermal time of 10 h, and a ratio of Ni(NO_3_)_2_·6H_2_O:Co(NO_3_)_2_·6H_2_O:thiourea of 1:2:6 were used. The hierarchical microstructure of pure NiCo_2_S_4_ was synthesized without graphene with a 3.17 g production.

### 3.3. Material Characterizations

The morphologies and elemental compositions were investigated using a field emission scanning electron microscope equipped with energy-dispersive X-ray (EDS) spectroscopy (FESEM, JEOL JSM-6700F, Tokyo, Japan). The crystal structure was determined by X-ray diffraction (XRD, Bruker D8 Discover, Karlsruhe, Germany) with CuKα irradiation (λ = 1.54184 Å) at 40 kV and 40 mA. Raman spectra were measured using 532 nm laser excitation on Raman microscopy (Raman, UniDRON, New Taipei City, Taiwan). The BET surface area and pore size were determined by the surface area and porosity analyzer (Micromeritics, ASAP-2020 Plus, Norcross, USA).

### 3.4. Experimental Design and Data Analysis

The central composite design (CCD)-based response surface methodology (RSM) was used to investigate the effect of preparation conditions on the electrochemical performance of the NCS@G hybrids. A central composite design with five levels, −1.682, −1, 0, +1, and +1.682, and a total of 20 runs, containing 6 axial points, 8 fractional factorial points, and 6 central points, was performed, as listed in Table 4. Real levels of independent variables were coded according to a specific Equation (4):Z = (Z_0_ − Z_c_)/∆Z (2)
where Z and Z_0_ represent the coded and real levels of the independent variable, respectively. ∆Z indicates step change while Z_c_ is the actual value at the central point. The actual values of independent variables were calculated according to the specific equation. Specific equations for G/NCS, hydrothermal time, and S/Ni are mentioned below Equations (5)–(7).
z_1_ = (G − 4)/2(3)
z_2_ = (T − 8)/2(4)
z_3_ = (S − 5)/1(5)
where G, T, and S represent the weight ratio of graphene to pure NCS powder, the hydrothermal time, and the molar ratio of CH_4_N_2_S/Ni(NO_3_)_2_·6H_2_O, respectively.

After the completion of the design of experiment, a second-order polynomial equation was used to indicate the responses as a function of independent variables, as follows (Equation (8)):*Y* = *a*_0_ + *a*_1_*X*_1_ + *a*_2_*X*_2_ + *a*_3_*X*_3_ + *a*_12_*X*_1_*X*_2_ + *a*_13_*X*_1_*X*_3_ + *a*_23_*X*_2_*X*_3_ + *a*_11_*X*_1_^2^ + *a*_22_*X*_2_^2^ + *a*_33_*X*_3_^2^(6)
where *Y* is the dependent variable (electrochemical performance of NCS@G/Ni composite electrode), *X*_1_, *X*_2_, and *X*_3_ are the corresponding independent parameters, such as the additional amount of graphene, the hydrothermal time, and a molar ratio of S/Ni, respectively. Additionally, *a*_0_ is the regression coefficient at the center point; *a*_1_–*a*_3_ are the linear coefficients; *a*_12_, *a*_13_, and *a*_23_ are the interaction coefficients, and *a*_11_, *a*_22_, and *a*_33_ are the quadratic coefficients. Experimental data of 20 runs were statistically analyzed using Design Expert Software (version 12). The statistical parameters such as lack-of-fit, predicted, adjusted correlation coefficients, and coefficient of variation were obtained to select the best fitting second-order polynomial model.

### 3.5. Preparation of NCS@G/Ni Electrodes and (G + AC)/Ni Electrodes

Without adding any conductive additives, 90 wt% NCS@G sample and 10 wt% polyvinylidene fluoride binder were mixed with a little N-methyl-2-pyrrolidinone solvent for homogeneity and then coated on the pre-cleaned Ni foam (3 × 2 cm^2^), and then dried at 80 °C for 8 h in a vacuum oven as an NCS@G/Ni composite electrode. The average mass loading of the final working electrode was approximately 2.0 mg cm^−2^.

A graphene dispersion with a concentration of 2.0 mg mL^−1^ was prepared by adding water to graphene paste. A certain amount of AC (ACS25) was added to the graphene dispersion as a mixed paste under continuous stirring. Then, the as-prepared mixed paste was coated uniformly on the pre-cleaned Ni foam without any adhesives or conductive agents. The coated (G + AC)/Ni was dried, measured, and weighed after each coating. Finally, a uniform film of (G + AC)/Ni electrode with graphene 1.35 (±0.02) and AC 0.3 (±0.01) mg cm^−2^ was obtained.

### 3.6. Electrochemical Measurements of the Single Electrode and Supercapattery Cell

All electrochemical measurements of the single electrode were carried out in 6 M KOH electrolyte. Electrochemical measurements of cyclic voltammogram (CV) and galvanostatic charge/discharge (GCD) testing were performed on an electrochemical workstation (6273E, CH Instruments Ins., Austin, USA) with the three-electrode system using the NCS@G/Ni as the working electrode; a Pt sheet and a standard calomel electrode (SCE) were used as the counter and reference electrodes. The EIS measurements were recorded in the 0.01 Hz to 100 kHz frequency range. From the GCD curves, the specific capacitance (C_m_) of the electrode was calculated based on the equation C_m_ = [(I × ∆t)/(m × V)], where I, ∆t, m, and V are the discharging current, the discharging time, the mass of the electroactive materials, and the voltage change, respectively.

A supercapattery cell was assembled with NCS@G/Ni and (G + AC)/Ni in 3.5 × 7 cm^2^ as the positive and negative electrodes in the 4M KOH-carboxymethyl cellulose (CMC) gel electrolyte. The as-fabricated supercapattery cell is denoted as NCS@G//(G + AC). The NCS@G/Ni and (G + AC)/Ni mass ratio was obtained based on the NCS@G/Ni and (G + AC)/Ni charge balance. The charge balance follows the relationship Q+ = Q−, and the optimized mass ratio of m_NCS@G_ to m_(G+AC)_ should be 0.5 in the supercapattery cell. A Battery Test System (CT3001A, Landt Instruments, Vestal, USA) was used to measure the cyclic stability of the supercapattery cell, where the voltage window was 0–1.6 V at 5 A·g^−1^.

The energy density, *E*, (Wh kg^−1^), and power density, *P* (W kg^−1^), are two key factors foe evaluating the practical utility of the supercapattery cells. The energy density and power density are calculated according to GCD curves using the following equations:(7)$$E=\frac{C_{S}\times \Delta V^{2}}{2\times 3.6}$$
(8)$$P=\frac{3600\times E}{\Delta t}$$

## 4. Conclusions

This study illustrates that response surface methodology is a powerful tool to optimize the hydrothermal synthesis of NCS coupled with graphene as the NCS@G/Ni composite electrodes. The ANOVA study indicates that the quadratic model is sufficient to describe and predict the responses of specific capacitance with changes in independent variables (G/NCS, S/Ni, and hydrothermal time). The results of specific capacitance showed that G/NCS and S/Ni have significant effects on the specific capacitance of NCS@G/Ni. The optimum condition was obtained through numerical optimization using the desirability function, and the optimized preparation conditions for the maximum specific capacitance of NCS@G (111)/Ni were G/NCS = 6.0%, hydrothermal time = 10.0, and S/Ni = 6.0. Due to the synergetic effect of nano-sized NCS and graphene with a meso/macropores porous structure, the NCS@G (111)/Ni displays a maximum specific capacitance of 2380 F g^−^^1^ at 5 A g^−^^1^. The fabricated NCS@G (111)//(G + AC) supercapattery cell delivered an energy density of 80 Wh kg^−^^1^ and a high power density of 4 kW kg^−^^1^. The cycling performance of the cell exhibits 75% capacitance after 5000 cycles at 2 A g^−^^1^. The synthesis strategy using RSM provides an effective way to minimize the processing and manufacturing costs of high-performance energy storage devices. 

## Data Availability

The data presented in this study supporting the results are available in the main text.

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
