# Peer review of "Response Surface Methodology Optimization in High-Performance Solid-State Supercapattery Cells Using NiCo2S4–Graphene Hybrids"

_molecules, 2022, doi:10.3390/molecules27206867_

Round 1

Reviewer 1 Report

This manuscript by Prof. Huang and Prof. Li et al. describes the employment of response surface methodology (RSM) to achieve the optimized synthetic recipe for the preparation of NiCo2S4-graphene hybrid materials. RSM analysis along with central composite design (CCD) has proven to be an effective tool to quickly evaluate a combination of several parameters for the optimized results. The author has shown that the hybrid material synthesized based on the prediction from the RSM analysis displays the experimental capacitance of 2376 F g-1 @5 A g-1. This observed value is highly consistent with the predicted one of 2317 F g-1, showing the usefulness of the RSM analytic tool.

There are some minor issues to be addressed by the author prior to the acceptance by the Journal. 

(1) Detailed experimental procedures for NCS@G(000) and NCS@G(111) should be included in the Experimental section.

(2) In section 3.6, “CMC” needs to be defined.

(3) Inconsistent font sizes are used in section 3.6.

(4) “ANOVA” needs to be defined.

(5) What is the difference between NCS@G(000) and NCS@G(111)? Even the former is claimed to possess the smaller pore size, there are no other data to show the inferiority of NCS@G(000) on capacitance performance.

(6) Why does NCS have such a small Rct? It is recommended to have the semicircle region of the Nyquist plots expanded, shown in the inset.

(7) In addition to PXRD, other characterization techniques are needed, such as EDX, XPS and CV? What is the actual composition of NCS on graphene material?

(8) To confirm NCS@G(111) having the best capacitance performance, the corresponding results of NCS@G(000) are required for comparison.

(9) The scope of content of this manuscript is better suited for the following journals: Batteries and Materials.

Reviewer 2 Report

This work describes the hydrothermal synthesis of NiCo2S4-graphene composite. The specific capacitance of the composite electrode was optimized using a combination of response surface approach and central composite design and compared to the conventional method. Overall, the findings of this work demonstrate that a response surface approach is a valuable tool for improving the composite's electrochemical properties of NiCo2S4 and graphene. However, the work needs to be revised in the ways described below before it can be accepted.

1-The introduction section needs to be further improved. For example, graphene/reduced graphene oxide's advantages were unclear. The authors only mentioned the ''high conductivity materials'' such as reduced graphene oxide or graphene.

Please cite these papers.

-  Electrochimica Acta 328 (2019) 135088.

-  Chemical Engineering Journal 409 (2021) 128216.

-  Molecules 2022, 27(12), 3696

2-The main text is a little confusing. In particular, the term ''NCS@G (111)/Ni'' indicates the growth of the material on Ni foam directly. In the Experimental Section, the authors stated that the materials were prepared by coating the pre-cleaned Ni foam.

3- In line 38, the authors stated, "to supply 50% of the world's energy demand by 2050 and eliminate fossil fuels in the future". Please, check the statement based on the reference.

4-The authors should correct the raw materials in scheme 1. They should also explain why the final materials contain Co3+ even though they started the reaction with Co2+.

5-Line 106. What did the authors mean by " will be adsorbed on the negative charge"?

6- In Fig. 1, how did the authors calculate the specific capacitance? The CV and GCD profiles are missing.

7-  Line 219. The JCPDS card no. 20-0782 of cubic phase NiCo2S4 differs from the JCPDS card in fig. 5.

8- In fig.6, the authors should calculate the ID/IG ratio to compare the two materials.

9- The author stated,'' It was obvious that the presence of graphene indeed could effectively increase the porosity and specific surface area''. The authors should explain this.

10- The high-frequency region in fig. 9 needs to zoom in and provide an equivalent electrical circuit model after fitting.

11-    Please check the energy and power density values.

12-   What is the typical reason for the sharp decline in stability from 100% to 75%? The authors should clarify this. 

13-  The authors should explain the low-rate capability of the supercapattery cell; the capability rate is 46%.

Round 2

Reviewer 2 Report

The manuscript was improved and can be published in the Molecules Journal. However, the authors should correct the I(D)/I(G) values in Figure 6. It is not matched with the text.

Author Response

We corrected I(D)/I(G) values in Figure 6 and please see the attachment.
